# History Lessons from the Late Joseon Dynasty Period of Korea: Human Technology (*Ondol*), Its Impacts on Forests and People, and the Role of the Government

**Jae Soo Bae** [1] **and Yeon-Su Kim** [2,*] 

1   Division of Forest Industry Research, National Institute of Forest Science, 57 Hoegi-ro, Cheongnyangri-dong, Dongdaemun-gu, Seoul 02455, Korea; forestory@korea.kr
2   School of Forestry, Northern Arizona University, P.O. Box 15018, Flagstaff, AZ 86011, USA
*   Correspondence: ysk@nau.edu; Tel.: +1-928-523-6643

**Abstract:** Historical analogies can help us contextualize new technical developments with social, cultural, and political forces at work. The late Joseon Dynasty period of Korea (1639–1910), a closed economy with detailed written records, provides a rare opportunity to examine a social-ecological system (SES) responding to drivers of change over a long period of time. Based on historical records and reconstructed data, we aim to: (1) characterize how the expansion of human technology, *Ondol* (traditional underfloor heating system), affected different subsystems and their interactions within the SES over time, (2) examine the role of the government in promoting the technology and regulating its impacts, and (3) summarize the pertinent lessons learned from old Korea for governing a modern-day bioeconomy. *Ondol* allows various forest biomass to be utilized as household fuel, including fuelwood, forest litter, and grass scraped from forest floor. Continuous biomass harvesting over 250 years to feed *Ondol* contributed to forest degradation and the forest ecosystem condition trapped in the early successional stage in the Korean Peninsula. The ecological changes were exacerbated by the Pine Policy with a singular focus on reserving Korean red pine (*Pinus densiflora* Siebold and Zucc.) for government uses. The policy failed to recognize basic needs of the public while countenancing an expansion of *Ondol* and a cultural preference for heated floors that propagated an increased use of biomass fuel. This case illustrates the importance of recognizing potential technology traps where a human innovation opened opportunities for more resource use. The lessons learned from old Korea show that bioeconomy transitions would require multifaceted governance responses while being cautious about being too closely tied to the dominant national agenda. Environmental history has much to offer for understanding the social and ecological systemic risks of the current technical developments. We call for more historical analogs from different parts of the world to "move forward by looking back".

**Keywords:** bioeconomy; forest history; Joseon Dynasty; Korea; *Ondol*; social-ecological system

## 1. Introduction

Transitioning to a bioeconomy, using renewable biological resources as feedstocks for energy generation and bio-based products has been one of the most prominent solutions for achieving sustainable development around the world [1]. Such a transition would require innovative governance responses to guide the transformation of a social-ecological system (SES) with technological advances, as well as the sustainable management of renewable resources [2,3]. Understanding forest landscapes as SESs can help us recognize not only social and ecological components within the system but, also, different spatial and temporal scales on which each component operates and their cross-scale

interactions [4]. There have been a number of historical analyses to understand long-term changes of forest landscapes as SESs [5–7]. The body of literature on the ecological and economic aspects of a bioeconomy has been also growing, although governing a bioeconomy as an SES has not received sufficient academic attention [8]. It is critical and urgent to understand how human technology affects different social and ecological components and their interactions within an SES [9,10]. Historical thinking and analogies can help us learn from history and contextualize current scientific and technical developments with the social, cultural, and political forces at work [11,12].

To contribute to our understanding of a bioeconomy as an SES, this paper examines the late period of the Joseon Dynasty of Korea (from the early 17th century to 1910), which provides a rare example of a closed economy with detailed written records, including daily reports and discussions in the royal court, as well as other publications. These historical records allow close examinations of interconnectedness and interactions among technology, social and biological processes, ecosystems, and the services that they provided. This period coincides with significant climate anomalies of reduced temperature, i.e., the "Little Ice Age". The population of Joseon also increased during this time. *Ondol* is a heating system in traditional Korean architecture that uses direct heat transfer from smoke generated by woodstoves or outdoor furnaces (*Agungi,* which allow fires for cooking to be used for heating but can also be constructed for heating purposes only) to heat the underside of the floor in the adjacent room (see Figure A1 for more detail on *Ondol*). Forest degradation of the Korean Peninsula before the Japanese occupation in 1910 has been frequently attributed by several historians to the "astronomical" amount of fuelwood demand due to *Ondol* [13–15]. The specific research objectives of this study are: (1) to characterize how the expansion of human technology, *Ondol*, affected different subsystems and their interactions within the SES over time, (2) to examine the role of the government in promoting the technology and regulating its impacts, and (3) to summarize pertinent lessons learned from old Korea for governing a modern-day bioeconomy.

This paper is organized as follows: Section 2 characterizes the SES of the late Joseon Dynasty period, in terms of the four core subsystems following the SES framework by Ostrom [16,17], while Section 3 presents the results based on our analysis of the historical documents. We conclude with salient lessons from old Korea for governing a modern-day SES.

## 2. Methods

### 2.1. Study Area—The Late Joseon Dynasty Period of Korea as an SES

Ostrom's general SES framework [16,17] is a highly useful tool for organizing different components within a system and their interactions, integrating diverse data from both natural and social sciences [18]. She proposed the four core subsystems: Resource Systems, Resource Units, Governance Systems, and Actors, with each subsystem containing multiple tiers with subdivisions of variables. These components interact, over temporal and spatial scales, with resulting outcomes, which then generate feedbacks to each of the components [16,17,19].

In this study, we characterized the late Joseon Dynasty period of Korea as an SES to organize the social and ecological components of the study area, which is defined both spatially (the Korean Peninsula, which was the Joseon's territory) and temporally (the late Joseon period from the early 17th century to 1910). The focal Resource Systems in this case are the forest ecosystems in Joseon's territory that contain Resource Units, such as different tree and plant species, biomass, and other resources. The focal Resource Units here are biomass fuel, including fuelwoods, forest litter, grass, and other biomass materials collected for burning. Governance Systems include Joseon's governing structure and the formal and informal rules that define property rights and forest uses. The focal Actors are households collecting forest biomass. We conceptualize that the human technology *Ondol* affected the mode and extents of interactions between Resource Units and Actors with two prominent drivers of change: the Little Ice Age and population growth (Figure 1).

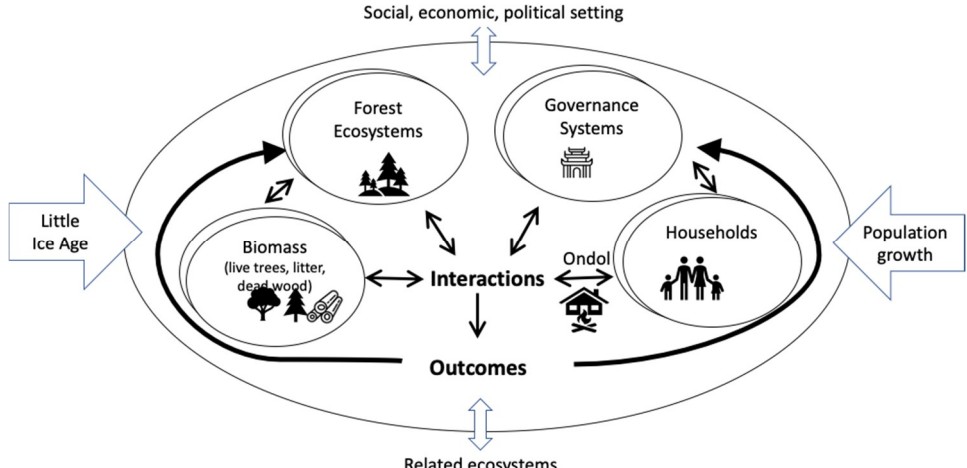

**Figure 1.** The late Joseon period as a social-ecological system (SES) with four core subsystems: (1) forest ecosystems within Joseon's territory as Resource Systems, (2) forest biomass for fuel as Resource Units, (3) Joseon's governing structure and formal and informal rules as Governance Systems, and (4) households as Actors.

### 2.1.1. Resource Systems and Units

The Korean Peninsula (approximately 22,000 km$^2$) has a temperate climate with four distinct seasons and an annual mean temperature ranging from 2.5 to 15 °C, but in January, the temperature can drop to −6 °C. Over half of all rainfall comes in the summer during the monsoon season, with an annual mean precipitation of 1000–1400 mm [20]. Except for the narrow subtropical belt along the southern coast, most areas contain deciduous broad-leaf and coniferous trees. More than 70% of the Korean Peninsula is mountainous with over 900 tree species. The most common trees species throughout the peninsula are various pine, especially Korean red pine (*Pinus densiflora* Siebold and Zucc.), and oak species, with a greater representation of conifer species in the north [21] and deciduous broad-leaf species in the south, as well as some evergreen species in the far south [22].

### 2.1.2. Governance Systems

The Joseon Dynasty (1392–1910) established its Constitution (*Gyeongguk Daejeon*) in 1397. Once subsequent revisions and additions were completed in 1485, it was never revised again and upheld as the set of governing principles for the nation until the kingdom's demise in 1910 with the colonization by Japan. The Constitution specified one of the governing principles as "Sharing nature's benefits among people" and banned the private possession of forests. Rural communities were expected to de facto manage and share forest resources through informal rules and cultural norms. Thus, no individual or institution other than the government was able to legally claim property rights over forestlands collectively or privately throughout the Joseon Dynasty for over 500 years. A common idiom for open access in Korean is *Mu Ju Gong San* (literally, "empty mountains with no owner"). Some scholars argued that the idiom is a result of mountainous forests having been commonly understood as open-access resources among the people of Joseon [23].

The late Joseon Dynasty period was heralded by devastating invasions by Japan (1592) and China (1636) that ravaged the whole kingdom. Post-war reconstruction and warships built to secure national defense sharply increased the timber demand, especially for large old-growth pine. The Joseon Dynasty had maintained a "closed economy" with limited trade with other countries and left no record of timber imports. Thus, both timber and biomass fuel demands had to be met by the limited supply from accessible forest resources within the Joseon's territory. While Korean households depended on forests for timber and a variety of non-timber forest products as food sources for centuries until

the rapid industrialization from the late-1960s onwards, the most essential resource from forests was fuelwood and other biomass to burn to survive the long, cold winter months [22,24].

### 2.1.3. Drivers of Change

The Little Ice Age: Although there are no accurate meteorological data, the kingdom of Joseon left detailed written records of reports and discussions in the royal court and daily affairs of the kings. These records reported incidents of unusual cold spells with increasing frequency from the 16th to the 18th centuries (peaking in the 17th century); for example, in 1655, the Eastern Sea froze in the spring, with cold weather and heavy snow in the far south subtropical region, and 900 government horses froze to death [15,25]. Between 1708 and 1709, vast areas around the eastern shore were reported as being frozen for over a month in the winter and again in the middle of the summer [15,25]. The term "Little Ice Age" refers to the period from the 16th to the 18th centuries with reduced temperatures, although the timing and nature of these cooler conditions varied from region to region (thus, not a globally synchronous Ice Age) [26–29]. Some argued that famines and mass migrations during the Little Ice Age drove the fall of the Ming Dynasty (1368 to 1644) in China [30,31]. Unusually low temperatures and increasingly unpredictable weather also had multiple effects on Joseon's society, with high frequencies of large-scale famines (the most notable in 1670–1671) and epidemic diseases, such as cycles of measles outbreaks every 10–20 years from the mid-17th century onward, as well as frequent smallpox epidemics [32]. Kim [25] even called the late Joseon period as "the era when disasters became part of life".

Population growth: Kwan and Shin [33] estimated the population size of the late Joseon period based on the records of the official population census conducted every three years from 1631 to 1861 with other records, including family tree books and provincial land registries. Although prone to undercounts due to the omission of women and children, as well as fraud (to avoid tax and drafts) [34], the record is still useful to demonstrate the longitudinal trend of the population change and to generate conservative estimates of the energy demand. The population in the late Joseon period increased over time, with periodic declines due to wars, epidemic diseases, and famines, followed by rapid recoveries (Figure 2). Other population estimates based on different methodologies, such as calculating the male population index based on family registers, show a similar overall picture of population growth during the late Joseon Dynasty, although the patterns of growth differ [35,36]. Some scholars attributed this general trend of population growth to the introduction of new rice cultivation techniques (i.e., transplanting seedings to expand the crop yield), the development of commerce, and the rise of merchant classes, which promoted the specialization and efficiency of agricultural production and manufacturing [37,38].

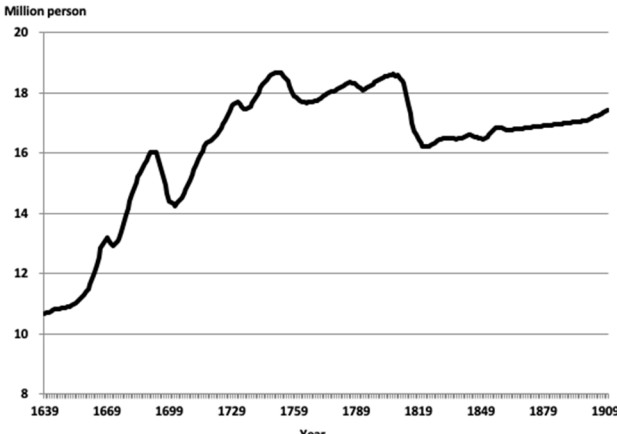

**Figure 2.** Population changes in the late Joseon Dynasty period from 1639 to 1910 (source: Kwan and Shin, 1977).

*2.2. Research Approach and Data Sources*

We analyzed various historical records, as well as contemporary literatures, to characterize the social and ecological subsystems in the late period of the Joseon Dynasty of Korea as an SES. The main data sources include: (1) the databases of forest-specific historical documents, maps, and forest inventory records that the National Institute of Forest Science, Republic of Korea, has been building in the last 20 years (Resource Systems and Units); (2) historical records, laws and regulations, and others classics translated by the Institute for the Translation of Korean Classics (Governance Systems and Actors); and (3) records of population and climate variations in the official census and archives of the royal court, as well as popular literature at the time (Drivers of Changes).

The historical records employed for our analysis included (1) The Annals of Joseon Dynasty (*Joseon Wangjo Sillok*), collections of daily reports and discussions in the royal court from 1413 to 1865 [39], (2) The Records of the Border Defense Council (*Bibyeonsa deungnok*), annual reports of the government agency (*Bibyeonsa*) that oversaw national security and internal affairs, including the protection and management of natural resources, from 1617 to 1892 [40], (3) Joseon Dynasty's Constitution (*Gyeongguk Daejeon*) [41] and other written records of laws and regulations for forest management (*Songgeumjeolmog*) [42], and (4) other historical and modern publications as cited.

## 3. Results

*3.1. Social and Cultural Impacts of Ondol on Households*

Various heating features resembling *Ondol* have existed in Korea since prehistorical times. *Ondol* was developed into the current form, which heats an entire floor with outside woodstoves, by the 11th century (Figure A1) [43]. The records of the royal court from the 15th and 16th centuries show that *Ondol* installation was typically limited to one or two rooms reserved for the old, the sick, or special guests [44]. *Ondol* became the defining feature of a Korean residence, from commoners' houses to the royal palace, during the 17th century. The records noting *Ondol* separately from other types of rooms (e.g., those with wooden floors) disappeared entirely by the early 18th century, when *Ondol* became synonymous with residential rooms [44]. A record from 1925 shows that, among the residents of the capital (current-day Seoul), 96.6% of Koreans reported that their residence had *Ondol* (compared to 24% among Japanese living in Seoul at the time) (the Dona-A Ilbo, 31 October 1925) [45]. *Ondol* expansion was initially prompted by the government (Seong's collections of current affairs published in 1790–1801, reprinted in 2006 [46]). After several forest fires in the early 17th century (most notably in 1672 [47]), government officials were encouraged to expand *Ondol* in their houses to collect and burn off surface fuel (e.g., pine needles). Chung [44] argued that the excessive level of pine needles noted in Seong's publications was because low temperatures during the Little Ice Age limited the nutrient recycling of organic matters. Frequent wars during this period allowed for the rapid expansion of *Ondol* in post-war reconstruction.

*Ondol* efficiently uses both radiative and convective heat transfer from fire without polluting the indoor air quality with smoke [43]. However, the lack of thermal insulation in traditional houses requires higher floor surface temperatures and encourages residents to sit or lie down to increase their body temperatures, shaping the unique residential culture of Korea [48]. *Ondol* is often referred to as the cultural womb of Korea, giving birth to a shared sense of culture and identity. The cultural significance of *Ondol* can be traced in idioms and expressions in the Korean language [49]. For example, being happy is often expressed as "having a warm backside and a full tummy" and seeking warmth as "wanting to be sizzled [on a hot floor]". The preference for *Ondol* and heated floors with high temperatures is still prevalent in modern Korea, as heated floors constitute normal residential architecture [43]. The Republic of Korea (ROK, South Korea) officially recognizes *Ondol* as a national cultural treasure (Number 135) [50].

*Ondol* and the increased fuel demand have been noted as drivers of forest degradation in Korea as early as the early 18th century. For example, Seo Yu Gu (1764–1845, renown scholar of the Rationalist

movement in 18th century Korea) wrote that "[t]he wasteful system of *Ondol* created three social menaces [ … ] A family of ten can spend 100 pieces of gold for fuelwood in one year. Small merchants and farmers are becoming impoverished by spending more than half of their income on fuelwood. This is the first menace. [...] There is not a single large tree in sight within 100 miles of cities. One cannot fulfil the duty for their parents by arranging a proper funeral [proper coffin]. This is the second menace [ … ] The mountains are bare, without stumps or dead roots. Landslides after heavy rains cover up crops and ruin harvests. This is the third menace" (The original publication date is unknown. A modern translation was published in 2016.) [51]. By the end of the Joseon Dynasty, completely rock-bare mountainsides became a common sight near cities, especially in the more populous central-southern areas (see Figure 3c). Some scholars attributed the Korean saying, "like woodstoves (*Agungi*) eating up mountains and still being hungry" to the massive biomass fuel demands to feed *Ondol* [52]. Others saw the Korean affinity for warmth and heated floors as a source of lethargy and complacency that led to the demise of the Joseon Dynasty, with some even saying that "*Ondol* ruined Korea" [53]. "Hygienically backward" housing, with *Ondol* allegedly encouraging lethargy, formed the bases for Japan's articulation of "Korean otherness" and for claiming cultural hegemony over the people of Joseon who were otherwise similar [54,55].

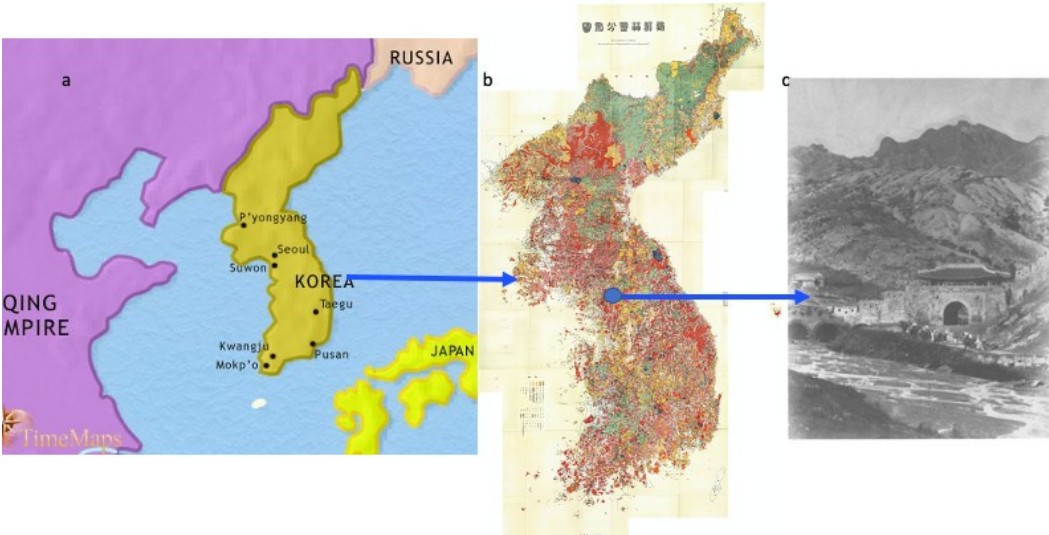

**Figure 3.** (**a**) A map of East Asia in the 17th century (source: Timemaps). (**b**) A map of Joseon in the year 1910 with the general tree species distribution observed and hand-drawn by the Japanese forest administrator, Saito (color code: red for Korean red pine (*Pinus densiflora* Siebold and Zucc.), green for other conifer species, and yellow for deciduous species; source: Bae and Kim, 2019). (**c**) A picture of Joseon's capital (current-day Seoul, Republic of Korea (ROK)) in the early 19th century (exact date unknown; source: Bae, 2002). Mountainous sides in the background with exposed bedrock show the extent of forest degradation.

## 3.2. Impacts of Ondol on Forest Ecosystems

Since there is no direct record of household fuel use in the late Joseon period, we postulate the extent of biomass fuel demand since the late 17th century based on a survey of household fuel use by the Japanese colonial government [56] and on the historical population estimates from 1639–1910 by Kwan and Shin [33]. According to the survey, the average household fuel demand for cooking (71%) and heating (29%) was estimated at a total of 7400 kg per household per year—37% forest litter (2739 kg), 30% firewood (2220 kg), 21% grass, and 12% agricultural by-products, such as grain husk [56]. To gauge the impacts of fuelwood demand on forest stock level, we converted the weight of fuelwood to volume (3.98 m$^3$ per household per year) based on the average ratio of common pine (455.81 kg/m$^3$) and oak (719.16 kg/m$^3$) species in the Korean Peninsula [57]. Historical records show that *Ondol*, which had been limited to one

or two rooms in residences of the ruling class, was expanded rapidly in the 17th century [44]. By the early 18th century, *Ondol* became commonly expected in bedrooms. We assumed the rate of *Ondol* use in the early 17th century to be 20%–40% of the 1911 estimates, with a gradual increase to 50%–70% by 1700. We constructed three difference scenarios proportional to the level in 1911 to estimate the extent of biomass fuel uses in the late Joseon Dynasty. Scenarios 1, 2, and 3 represent low, medium, and high levels of Ondol adoption at 20%, 30%, and 40% of the rate in 1911 (See Table 1).

**Table 1.** Scenarios with different *Ondol* adoption rates.

| Scenarios | *Ondol* Adoption Rate (%) | | | |
| --- | --- | --- | --- | --- |
| | 1639 | 1639–1700 | 1700–1800 | 1800–1911 |
| Scenario 1: Low | 20 | 20–50 | 50–90 | 90–100 |
| Scenario 2: Medium | 30 | 30–60 | 60–90 | 90–100 |
| Scenario 3: High | 40 | 40–70 | 70–90 | 90–100 |

Population estimates are based on those of Kwan and Shin [33], assuming an average household size of 4.88. Household fuel uses were estimated, based on the 1911 survey [56], at 2739 kg of forest litter and 3.98 m$^3$ of fuelwood per household per year as of 1910. The estimated fuelwood demands under the three scenarios are presented in Table 2 for fuelwood and Table 3 for forest litter. The annual average of forest biomass fuel demand from 1639–1910 is estimated to be 9.9~10.9 million m$^3$ of fuelwood and 6.5–7.5 million tons of forest litter (see Tables 2 and 3).

**Table 2.** Fuelwood demand by scenario (unit: million m$^3$).

| Scenarios | 1639 | 1910 | 1639–1910 | |
| --- | --- | --- | --- | --- |
| | | | Cumulative Fuel Demand | Annual Average |
| Scenario 1: Low | 1.74 | 14.2 | 2694 | 9.94 |
| Scenario 2: Medium | 2.61 | 14.2 | 2828 | 10.43 |
| Scenario 3: High | 3.48 | 14.2 | 2962 | 10.93 |

**Table 3.** Forest litter demand by scenario (unit: million ton).

| Scenarios | 1639 | 1910 | 1639–1910 | |
| --- | --- | --- | --- | --- |
| | | | Cumulative Fuel Demand | Annual Average |
| Scenario 1: Low | 1.20 | 9.77 | 1749 | 6.46 |
| Scenario 2: Medium | 1.80 | 9.77 | 1945 | 7.18 |
| Scenario 3: High | 2.40 | 9.77 | 2089 | 7.71 |

To estimate the additional biomass fuel demand created by *Ondol*, we used the adoption rate in 1639 as the baseline, as this would have been the case if *Ondol* continued to be in limited use for one or two rooms in residences of the ruling class. Additional fuelwood and forest litter demands are presented in Tables 4 and 5. The forest biomass fuel demand increase due to the expansion of *Ondol* is estimated to translate into approximately 5.6–7.3 million m$^3$ of fuelwood and 3.9–5 million tons of forest litter.

**Table 4.** Additional fuelwood demand by scenario (unit: million m³).

| Scenarios | Cumulative Fuel Demand | | Total Additional Demand (A–B) | Annual Average of Additional Demand |
|---|---|---|---|---|
| | Baseline (A) | *Ondol* Expansion (B) | | |
| Scenario 1: Low | 721 | 2694 | 1973 | 7.28 |
| Scenario 2: Medium | 1082 | 2828 | 1746 | 6.44 |
| Scenario 3: High | 1442 | 2962 | 1520 | 5.61 |

**Table 5.** Additional forest litter demand by scenario (unit: million ton).

| Scenarios | Cumulative Fuel Demand | | Total Additional Demand (A–B) | Annual Average of Additional Demand |
|---|---|---|---|---|
| | Baseline (A) | *Ondol* Expansion (B) | | |
| Scenario 1: Low | 496 | 1749 | 1253 | 4.62 |
| Scenario 2: Medium | 744 | 1945 | 1201 | 4.43 |
| Scenario 3: High | 992 | 2089 | 1097 | 4.05 |

The extended biomass fuel demand had two main ecological consequences: forest degradation, especially around populous areas, and a forest ecosystem condition trapped in an early successional stage. The first modern record of the forest stock level shows 275 million m³, with a mean growing stock (MGS) of 17 m³/ha as of 1927 [58]. Forest stock levels in current-day Korea total 1237 million m³: 319 million m³ with an MGS of 63 m³/ha for North Korea and 918 million m³ with an MGS of 148 m³/ha for South Korea [59]. The forest stock levels of current-day South Korea represent the upper range of the growing potential of forests, considering its remarkable forest transition history in the modern era [24]. Since Japan colonized the Korean Peninsula in 1910, massive logging was done to feed Japan's war efforts, especially in the northern provinces bordering China. Even after accounting for the forest degradation from 1910 to 1927, there is little doubt that the forest stock level was very low by the end of the Joseon Dynasty. To understand the magnitude of the fuelwood demand and its impacts, we assume that the forest stock level in the 16th century was approximately twice that in 1927 (~500 million m³) after the two devastating foreign invasions (in 1592 and 1636). The mean growing stock level would be approximately 15 million m³ per year, with a 3% natural growth rate. This means that household demands for fuelwood alone consumed 60%–70% of the total annual growth of forests, even without accounting for the timber demands of buildings and ships and other nondomestic fuelwood demands for, e.g., pottery making and blacksmithing. However, the pressures on forest biomass would not have been evenly distributed throughout the Korean Peninsula, as the lands with a lower elevation and a gentler slope in the south had been historically more densely populated (Figure 4). Fuelwood harvests would have exceeded the annual growing stock level in the forests near densely populated areas, resulting in forest degradation (Figure 4d).

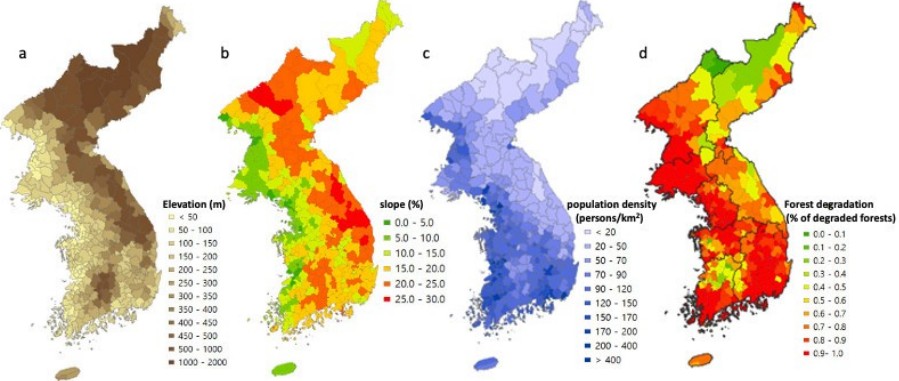

**Figure 4.** By district, as of 1910: (**a**) average elevation, (**b**) average slope, (**c**) population density/km², and (**d**) deforestation rate (source: Bae and Kim, 2019).

Unlike other heating methods generating direct heat from indoor fireplaces (e.g., Europe) or stoves (e.g., Japan), *Ondol* allow for an extensive use of inefficient fuels with high amounts of soot and smoke, such as forest litter, grass, and agricultural by-products, for heating. The continuous and widespread collection of forest litter caused by scraping the forest floor profoundly affected the forest ecosystem in the Korea Peninsula over time through changes in depth, organic matter, and soil chemistry, leading to changes in the forest structure and composition [60]. These slow-moving changes are hard to detect over the short term, even though they control the dynamics of fast-moving variables, such as tree species, the level of biomass, and other ecosystem services, which are often the focus of management [61]. Mismatches of spatial and temporal changes in ecological processes and human responses when governing forest landscapes may generate unintended consequences in forest management [4]. Based on the digital reconstruction of hand-drawn maps and other historical records left by a Japanese colonial forest administrator named Saito, Bae and Kim [62] argued that Korean red pine (*Pinus densiflora*) was the dominating tree species across the Korean Peninsula in 1910. Pine-dominated forests were observed to comprise 43% of the total forest area and 78% of the forests in the south-central regions (current-day ROK) (Figure 5a).

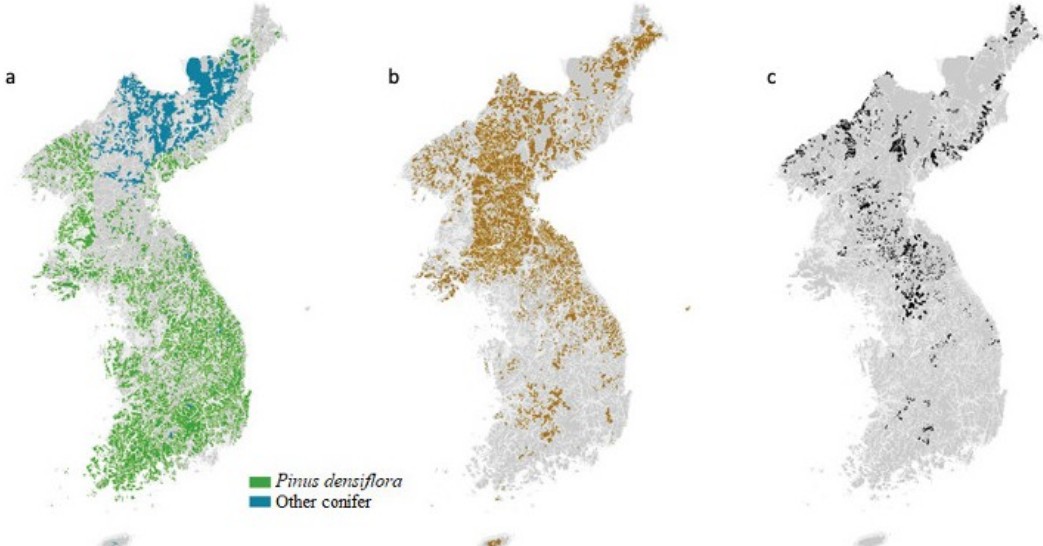

**Figure 5.** Distribution of forests (total 15.7 million ha) by the dominant species observed by Saito in 1910: (**a**) coniferous forests (41% of total forest land, 30% of which is forest dominated by Korean red pine (*Pinus densiflora)*, shown in green), (**b**) deciduous forests (30%), and (**c**) slash and burn Figure 3. (source: Bae and Kim, 2019).

The mountainous peaks in the Korean Peninsula rarely exceed 1500 m, but they are common throughout the landscapes and provide diverse microclimates along the elevational gradients and climate zones. Common hardwood species, such as various oak species, are shade-tolerant and require moist layers of organic duff to geminate, thus frequently occurring in the late-successional stage [63]. Korean red pine (*Pinus densiflora*) is one of the early successional species that are drought-tolerant and can germinate under a full sun and under poor soil conditions [64]. It is not competitive with oak species in good soil and moisture conditions [64] and is thus expected to persist in the mountain ridges and steep slopes [60,65]. The proliferation and dominance of Korean red pine (*Pinus densiflora*) throughout the peninsula, especially around the densely populated south-central regions, can be attributed to the anthropogenic pressures driving multiple slow-moving and system-wide changes in the forest ecosystems. First, when the forests are continuously harvested for all matters of biomass, from over- to understories, including mature and young trees, shrubs, saplings, grass, and forest litter, soil surfaces are exposed to full sun, and residual fertility declines, preventing the forest

ecosystem from transitioning to later successional stages. This is evident by the relative dominance of other conifer and hardwood species in the northern region, where rugged terrains provide limited access to forests, and the population density was low (Figures 4c and 5). A previous study noted that the areas of pine-dominated forests have continuously decreased in recent years in ROK with absences of anthropogenic pressures [66]. Second, biomass fuel demands combined with a policy favoring pine, which is discussed in the next section, lowered the level of ecosystem-wide biodiversity. The prevalence of pine-dominated overstories results in a low biodiversity of understory flora and fauna and in an increased vulnerability to natural disturbances, such as fires, insects, and disease outbreaks. The continuous scraping of forest floors to collect biomass made soil surfaces susceptible to runoffs and soil erosions, resulting in a decrease in topsoil depth that ultimately exposed underlying bedrocks (as shown in the historical picture in Figure 3c).

These ecological changes were exacerbated by other contributing factors, such as wars, slash and burn agricultures, and charcoal production. As noted above, the late Joseon period is demarcated after two major wars in 1592 and 1636. There was no major war during this period. The impacts of internal conflicts and small-scale invasions on forests would have been relatively limited and sporadic, compared to the cumulative impacts of continuously increasing fuel demands by population growth and the expansion of *Ondol* over 250 years. The impacts of land use change pressure on forests were limited to slash and burn agriculture, which accounted for 3% of forests in 1910 (Figure 5c) [62]. *Ondol* enabled households to use inefficient fuels, such as forest and agricultural biomass, while avoiding indoor air pollution. Charcoals were rarely used by common households, accounting for only 2% of fuel sources in 1910 [56]. Their uses were limited to the royal palace and residences of the very rich [67].

Over time, these ecological changes also resulted in profound changes in human preferences and culture. For example, Korean red pine (*Pinus densiflora)* is simply called *Sonamu* in the Korean language (literally "pine tree") and is a national symbol representing unbending loyalty, even featured in the national anthem. *Sonamu* has been consistently selected as the favorite tree among the public of South Korea in the Gallup-Korea surveys from 1991 to 2015, chosen by 45.7% (1997) to 67.7% (2015) of respondents [68].

### 3.3. The Role of Governance Systems

The Joseon Dynasty existed between two geopolitical superpowers throughout its existence and experienced numerous invasions since its inception. However, invasions by Japan became more frequent and intense over time, which made defending the southern coasts facing Japan a national priority [69]. There are multiple historical records specifying large old-growth pines over 100 years old as the materials used to build warships since the early Joseon Dynasty period (e.g., the records of the royal court in 1419 and in 1430) [39]. Japanese ships at the time were often constructed with Japanese cedar (*Cryptomeria japonica* (L. f.) D. Don) or Japanese fir (*Abies firma* Siebold and Zucc.) built for speed, while Joseon pursued building smaller but sturdier ships with old-growth pine, as the species can produce large, high-quality lumber near shipyards in coastal areas [70]. However, lumber from old-growth pine was also an important material for constructing buildings, coffins (as large funerals and extended periods of mourning are important in Korean culture), and ships for commerce and transportation, in addition to warships. The resource conflicts over old-growth pine were evident in as early as 1418, when the royal court heard the argument for limiting pine use for constructing government buildings and private residences (the Annals of the Joseon Dynasty, 27 August 1448) [39]. However, conflicts over forest resources were not apparent until the 17th century, when the "Pine Policy" (*Song Jeong*, literally "pine policy") increasingly became a regular point of discussion in the royal court [71]. By the late 17th century, the Joseon Dynasty established 635 forest reserves, mostly reserved for providing large, high-quality timber for warship construction but, also, including 60 locations reserved for providing materials for making coffins for royalty and other official uses. As increasing and competing demands for pine intensified over time, the royal court (the central government) started

developing more regulations and distributed detailed guidelines for implementing the Pine Policy to government agencies and provinces (e.g., in 1684 and 1788) [72].

As stated clearly in its name, the Pine Policy singularly focused on the protection of Korean red pine (*Pinus densiflora*) through the forest reserve system to secure timber for building warships. The rising timber demand for other government uses had to be met by the forests outside the reserves. Pine harvesting was increasingly prohibited in all forests in the later years, as noted in the Records of the Border Defense Council, 7 February 1753 [40]. This control of timber resources also created bureaucratic power struggles between the National Defense Agency and provincial governments. Initially, the management authority of forest reserves was allocated to low-ranking officials with national defense responsibility reporting to the provincial governor. However, forest fires in 1762 and subsequent debates for clarifying the responsible authorities prompted the king to clearly demarcate the authority to the Marine Security Agency, stating that "protecting pine is a matter of national security". In the earlier Joseon period, there were multiple cases in which the government, to alleviate timber or firewood shortages, allowed some harvesting in the forest reserves. However, framing the Pine Policy under national security in the later period led to a boost of bureaucratic power of the Marine Security Agency and their local officials to protect and control timber from the forest reserves. Around the same time (1750), the resources and budget allocated to the Marine Security Agency was reduced by 40% when the royal court decided to remove emergency resources and funds allocated to the agency during the war of 1592 [73]. The officials in the agency sought to mitigate the gap by exercising their authority over the forest reserves by collecting fines and other penalties for the alleged illegal harvesting of pine from the forest reserves [72,74]. For example, one governor in 1798 reported incidents of the Marine Security Agency frequently harassing the public for any kinds of pine in their possessions, and the public was afraid to be seen even with pine branches, delaying constructions and funerals [74]. The need for reserving forests for warship construction allowed the bureaucracy to exploit the structural power of the rigid protection policies, which undermined the legitimacy and effectiveness of the Pine Policy overall.

As increasing and competing demands for pine intensified over time, the central government also tried to reduce the demands for pine by extending the periods between repairs of warships and adopting new technologies (e.g., replacing wooden nails with iron ones), as well as limiting the number and sizes of new constructions of government buildings, as noted in the Records of the Border Defense Council in 1787 [40]. However, these policies for reducing demands had little impact, because they provided no incentives for local officials who oversaw shipyards to reduce their requests of timber. The royal court heard multiple cases of timber resources being wasted in shipyards, either by overreporting the number of ships to be rebuilt or by overestimating the timber needed for building ships, which was suspected to be intentional to sell off "leftover" timber (the Annals of the Joseon Dynasty, 1770, 1781, and 1800) [39]. With their revenue directly linked to how much timber was being harvested in each year from the forest reserves, the Marine Security Agency also had no motivation to question the quantity of timber requested by the shipyards.

The government not only determines the social distribution of forest resources (structural power) but, also, intentionally and unintentionally promotes human technology by affecting people's preferences and culture (ideational power) [10]. The government initially encouraged *Ondol* expansion to burn off excessive forest fuel built up in the early 17th century, as described above. The late Joseon period is also marked by increasingly complex bureaucratic structures and power struggles [75]. Sprawling new government buildings were built with *Ondol* even in servants' quarters (the Annals of the Joseon Dynasty, 5 March 1624) [39], which contributed to the fuel shortage and rapidly rising prices of fuel. By the late 18th century, the Joseon Dynasty was spending 18%–19% of the government budget on fuelwoods and charcoal [76]. Expansive uses of *Ondol* by the ruling class did nothing to dampen the growing resentments among the commoners, further eroding the public acceptance of the Pine Policy. The ideational power of the governance system was exercised by the intentional promotion of *Ondol*, as well as unintentional normalization, through its extensive uses in government buildings.

## 4. Discussion

When the Japanese forest administrator Saito arrived in Korea in 1910, one of his first observations was the open-access situation driving forest degradation [77]. Poor forest conditions were later framed by the colonial government as evidence for the irresponsible nature of Koreans and a rationale for Japanese occupation and closing the common [55]. However, failures are common throughout human history for balancing fast-growing demands for forest resources responding to the drivers of changes with renewable, yet slow-growing and limited, supplies from the forest ecosystem [4,78]. The lessons from the case of the SES that evolved over 250 years are salient for understanding how human technologies can affect the resource dynamics of an SES. We summarize four main lessons learned from old Korea below.

First, technological innovations may open opportunities for more efficient, but not necessarily less, resource use. Jevon's paradox occurs when the increase in energy efficiency per unit leads to an overall higher level of production and energy demand and more resource competition [79–84]. While highly active in promoting new technologies, most states promoting the bioeconomy do not recognize the potentially negative consequences of bioenergy development beyond food security linked to land and water resources, which include inequality and poverty, climate, or health risks [1]. If the adoption of new technologies increases the inequality and policies ignore the needs of the poor, unintentional technological traps can intensify conflicts over resources and drive slow-moving yet profound changes in Resource Systems. The case of *Ondol* illustrates the importance of recognizing a range of social and ecological issues when promoting new technologies.

Second, a single policy instrument, such as the Pine Policy, is rarely effective in addressing multifaceted issues related to bioenergy development [85,86]. Joseon's policy offered limited immediate benefits to their constituencies living through the Little Ice Age and provided little motivation and incentives for bureaucracies at different levels to exceed policy requirements and explore alternatives for reducing timber and fuelwood demands. The concentrated decision-making power in the royal court (the central government) limited the engagement of provincial and local-level institutions. The policy was ultimately exploited by the bureaucrats to expand their power and secure resources, which brewed resentments among the public, perceiving the policy as self-serving of the ruling class.

Third, governance responses focusing on a single species, even with native species, present high risks of unintended ecosystem-wide consequences. With growing global enthusiasm for a bioeconomy, the search for high-yielding crops thriving in marginal conditions is intensifying, as well as the warning for the invasive potential of such crops [87]. As seen in this study, social and ecological changes may not be detected in the short term, as the controlling variables, such as soil and landscape conditions, climate, and culture, change slowly, typically over centuries [88]. Ecological changes occurring through disruptions of social systems, such as wars, rapid industrialization, and changing populations, may also create a discrepancy between human perceptions and ecological realities [89,90].

Fourth, the Pine Policy framed under the national security agenda allowed the design of policy instruments and implementation to be inflexible to changing demands and evolving contexts of climate and population changes. The energy policy discourse is still often linked to the core national agenda, such as national security. Many scholars have argued that a low-carbon economy cannot be realized without reframing energy policies either to be more tightly linked to the dominant national agenda or to override it [91]. Although it is hard to directly compare the kingdom of Joseon with modern democratic societies, promoting a bioeconomy by linking the need for national security may be a risky proposition considering the rigidity of the policy that such framing may invite.

## 5. Conclusions

The lessons learned from the late Joseon Dynasty period of Korea are not just a cautionary tale of failed resource management that happened someplace else long ago. The case provided illustrative examples of known concepts, such as Jevon's paradox and wicked problems of resource management,

as well as the new insights on the drawbacks of rigid policy framing and the role of the government in shaping human preference and culture over time.

We face the Anthropocene age, where human dominance over the climate and the environment is increasing, with multiple intertwined forces at work driving ecological and social changes [92]. The need for new and reformed institutions in multiple scales is critical now more than ever, although institutions still tend to focus on single problems and ignore system-wide interactions [93]. We argue that the field of environmental history has much to offer for understanding the social and ecological systemic risks of the current scientific and technical developments. However, historical datasets recording environmental changes outside the Western world were often constructed with colonial narratives [11]. Examining long-term changes of the human-nature relationship based on such datasets would require critical inspections of how and why they were constructed [11]. Employing the system's perspective in historical analyses of environmental changes can help discern social, economic, and political forces at work with ecological changes and place the lessons learned in the contexts of contemporary bioeconomy developments. We call for more historical analogs from different parts of the world to "move forward by looking back".

**Author Contributions:** Conceptualization, J.S.B.; methodology, J.S.B. and Y.-S.K.; formal analysis, J.S.B. and Y.-S.K.; investigation, J.S.B.; resources and data curation, J.S.B.; writing—original draft preparation, J.S.B. and Y.-S.K.; writing—review and editing, J.S.B. and Y.-S.K.; visualization, J.S.B. and Y.-S.K.; and project administration, J.S.B. All authors have read and agreed to the published version of the manuscript.

**Funding:** This research received no external funding.

**Acknowledgments:** We thank the administrative and technical support from the research team at the Forest Industry Division, the National Institute of Forest Science, Republic of Korea. We appreciate the comments made by Dave Egan on the early version of the paper, as well as those by two anonymous reviewers.

**Conflicts of Interest:** The authors declare that there are no conflicts of interest.

## Appendix A  Traditional *Ondol*

The traditional *Ondol* allowed cooking fires in the morning and evening to heat a space by storing heat in the stone or clay underneath the floor. The figure below shows the traditional *Ondol* structure.

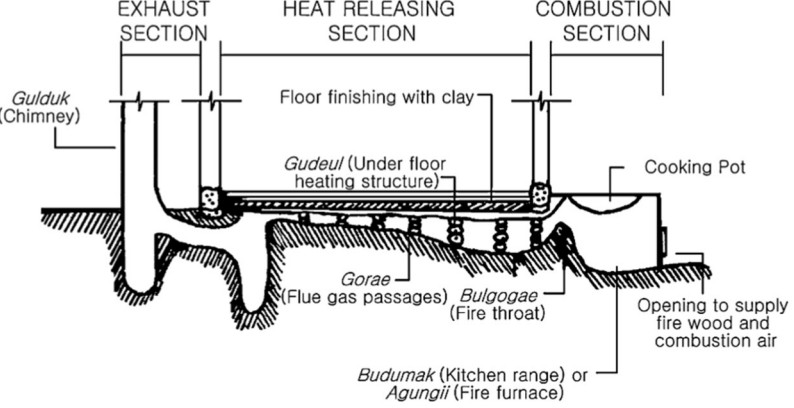

**Figure A1.** Structure of the traditional *Ondol* (source: Yeo et al., 2003).

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
