# Peer review of "History Lessons from the Late Joseon Dynasty Period of Korea: Human Technology (Ondol), Its Impacts on Forests and People, and the Role of the Government"

_forests, doi:10.3390/f11121314_

Round 1

Reviewer 1 Report

Please include more key elements from your discussion (4 points) and conclusions in the abstract.

Please describe the research approach/method more broadly and in detail (in addition to theoretical framework and materials).

Please consider including more recommendations and lessons learnt in the conclusions and maybe future research as well. You could address sustainability of bioeconomy and circular economy aspects as well.

Interesting and well written paper.

Author Response

Thanks for your time and effort for providing the review. We appreciate the comments and addressed all of them to the full extent possible. 

Attached is our point-by-pont responses for both reviewers' comments.

Reviewer 2 Report

This manuscript was described the forest resource use in Korea from the Late Joseon Dynasty and I think it is quite interesting story for considering Korean forest history.

When reading the title and abstract in this manuscript, I recognized that the authors may be focused on for discussing t he relationship between Ondol and Forests. But when I read the main text, the contents of manuscript did not match this title and abstract. The authors should revised title and abstract in this manuscript.

In addition, this manuscript should be developed several points. Please develop the manuscript based on the comments described as bellows.

Introduction parts

Please develop background and problem statements in this manuscript. In this article, it is very weak justification why the authors want to discuss ecological and economic aspects of the bioeconomy.  I think the authors should argue more their research interest and set up problem statements more clearly.

Materials and Methods

The description of this section did not fit the material and method. The authors just explained the Joseon Dynasty of Korea mainly. Please develop here to explain the materials and method in this parts.

Line 223 and 231

The location of all tables should be moved to near text where the authors argued these points, not appendix.

Line 303

Did the war really less impact to deforestation in Korea? As the authors mentioned while Japan occupy Korea, it had heavy deforestation occurred in line 46-47.

Line 306

Charcoal using was very limited, but what about fuel using by local people for Ondol?

References

 The writing style of references should be followed “the instruction for author”. Please check the contents in your references and revise these following “the instruction for author” . Please check my comments in the attached file. There are still many points to be revised.

Author Response

(The authors gave the same response as above.)

Round 2

Reviewer 2 Report

Dear authors,

 Based on the revised manuscript, I recognized that authors could successfully revised it.

I have small comments for writing style of references. See attached file.

Author Response

Thanks for the careful review of the list of references. All errors in the citations have been corrected and proof-read by MDPI copy editor.